

# Leeches *Baicalobdella torquata* feed on hemolymph but have a low effect on the cellular immune response of amphipod *Eulimnogammarus verrucosus* from Lake Baikal

Anna Nazarova[1], Andrei Mutin[1], Denis Skafar[2,3], Nadezhda Bolbat[1], Sofya Sedova[1], Polina Chupalova[1], Vasiliy Pomazkin[1], Polina Drozdova[1,4,5], Anton Gurkov[1,4] and Maxim Timofeyev[1]

[1] Institute of Biology, Irkutsk State University, Irkutsk, Russia
[2] Faculty of Biology, Department of Aquatic Bioresources and Aquaculture, Kuban State University, Krasnodar, Russia
[3] Krasnodar Department, Azov Estuaries Sector, Azov-Black Sea Branch of the Russian Federal Research Institute of Fisheries and Oceanography, Rostov-on-Don, Russia
[4] Baikal Research Centre, Irkutsk, Russia
[5] Faculty of Biology and Soil Sciences, Irkutsk State University, Irkutsk, Russia

Corresponding author
Maxim Timofeyev,
m.a.timofeyev@gmail.com

## ABSTRACT

Lake Baikal is one of the largest and oldest freshwater reservoirs on the planet with a huge endemic diversity of amphipods (Amphipoda, Crustacea). These crustaceans have various symbiotic relationships, including the rarely described phenomenon of leech parasitism on amphipods. It is known that leeches feeding on hemolymph of crustacean hosts can influence their physiology, especially under stressful conditions. Here we show that leeches *Baicalobdella torquata* (Grube, 1871) found on gills of *Eulimnogammarus verrucosus* (Gerstfeldt, 1858), one of the most abundant amphipods in the Baikal littoral zone, indeed feed on the hemolymph of their host. However, the leech infection had no effect on immune parameters such as hemocyte concentration or phenoloxidase activity and also did not affect glycogen content. The intensity of hemocyte reaction to foreign bodies in a primary culture was identical between leech-free and leech-infected animals. Artificial infection with leeches also had only a subtle effect on the course of a model microbial infection in terms of hemocyte concentration and composition. Despite we cannot fully exclude deleterious effects of the parasites, our study indicates a low influence of a few leeches on *E. verrucosus* and shows that leech-infected amphipods can be used at least for some types of ecophysiological experiments.

## INTRODUCTION

Various parasites are now considered as a significant environmental factor influencing the survival of aquatic animals under stressful conditions and sometimes acting synergistically
with such factors as pollution (*Sures, 2006*; *Grabner, Rothe & Sures, 2023a*; *Öktener & Bănăduc, 2023*). In particular, some parasites have been shown to manipulate behavior, distort sex ratios, modify energy budgets and compromise the immune defense in amphipods (Amphipoda, Crustacea), one of the most important groups of freshwater invertebrates (*Giari et al., 2020*).

Leeches are annelid worms (Hirudinea, Annelida), many species of which parasitize various animals and feed on the host blood or hemolymph. Importantly, saliva components of these parasites can have anticoagulant, anti-inflammatory, and other roles, but such bioactive components and their effects are mostly studied in medically important species (*Salzet, 2001*; *Zaidi et al., 2011*; *Liu et al., 2019*). Leeches and crustaceans can exist in different types of ecological relationships. For example, leeches of the species *Myzobdella lugubris* Leidy, 1851 are parasites of crabs *Callinectes bocourti* Milne-Edwards, 1879 feeding on their hemolymph and laying eggs on the surface of the crab body (*Zara et al., 2009*). The South African leech *Marsupiobdella africana* is a facultative ectoparasite of the amphibian *Xenopus laevis* and has a phoretic relationship (*i.e.,* promoting spreading of the attached phoront) with the freshwater crab *Potamonautes perlatus* Milne-Edwards, 1837. The sex of the host crab has been shown to be important for leech infestation. In addition, the period of residence of the leeches on crabs coincides with the development of leech eggs, which may indicate additional benefits of these relationships for leeches (*Badets & Preez, 2014*). The crayfish *Orconectes rusticus* (Girard, 1852) has the cleaning leech-like symbiont *Cambarincola fallax* Hoffman, 1963 that removes fouling organisms and thus improves growth rates of the host (*Brown, Creed & Dobson, 2002*; *Keller, 1992*; *Lee, Kim & Choe, 2009*). The fish leech *Johanssonia arctica* (Johansson, 1898) is also an epibiont of the red king crab *Paralithodes camtschaticus* (Tilesius, 1815) (*Dvoretsky & Dvoretsky, 2021*; *Dvoretsky & Dvoretsky, 2009*).

Lake Baikal is among the largest and most ancient freshwater reservoirs on the planet and also the birthplace of an outstanding endemic diversity of amphipods playing various roles in the lake ecosystem (*Brown et al., 2021*). Over 350 morphological species and subspecies of amphipods have been described from Baikal, constituting about 19% of all known freshwater species and demonstrating tremendous morphological variety (*Väinölä et al., 2008*; *Takhteev, Berezina & Sidorov, 2015*). Yet, symbionts and parasites of Baikal amphipods and their potential influence on the physiology of these crustaceans are understudied. It is known that the hemolymph of the amphipods can contain various bacteria (*Shchapova et al., 2021*) and DNA of microsporidians (*Dimova et al., 2018*). Despite the relatively low numbers of analyzed animals, those studies suggest that the fraction of individuals with detectable microsporidian DNA is generally on the order of percents, while the infection rate with live bacteria can be as high as 80%. Baikal endemic amphipods are also known to be intermediate hosts for acanthocephalans, but the fraction of infected individuals is generally low (*Baldanova & Pronin, 2001*).

However, the parasites that can be most easily found on amphipods in Lake Baikal are leeches. According to our observations, leeches are mostly attached to the gills of the largest morphological species in the Baikal littoral zone, such as *Eulimnogammarus verrucosus* (Gerstfeldt, 1858) or *Pallasea cancellus* (Pallas, 1772) and much less often to

a smaller *E. vittatus* (Dybowsky, 1874). The hypothesis that the parasites prefer larger species as hosts is also supported by observations of leeches on even larger deep-water Baikal amphipods (*Kaygorodova, 2015*). Again, according to our preliminary observations in *E. verrucosus*, leeches can infect a substantial proportion of the population on the order of dozens of percents at least in some seasons. These parasites of *E. verrucosus* belong to the genus *Baicalobdella* containing at least two species, *B. cottidarum* Dogiel, 1957 and *B. torquata* (Grube, 1871) (*Lukin, 1976*; *Bauer, 1987*; *Timoshkin, 2001*). *E. verrucosus* is a widespread and abundant morphological species in the littoral zone of Lake Baikal (*Gurkov et al., 2019*), and yet the influence of leeches on its physiology is fairly unstudied. Moreover, the whole phenomenon of leeches infecting amphipods seems to be very rare, if not unique to Lake Baikal, which might be related to the larger size of many Baikal endemics in comparison to most freshwater amphipods. A literature search gave us no other examples of such a phenomenon, and a recent review categorizing parasites of amphipods does not mention leeches at all (*Bojko & Ovcharenko, 2019*).

If leeches indeed feed on the hemolymph of amphipods in Lake Baikal (*i.e.,* if they are not just phoronts), the infection may directly impair amphipod immune defense and indirectly lower the available energy resources besides the potential effects of leech saliva. The crustacean immune system relies on hemolymph components such as hemocytes (*i.e.,* circulating cells) and the phenoloxidase system. Hemocytes perform phagocytosis and encapsulation of foreign bodies, while phenoloxidase is responsible for the melanization process, which is also a part of foreign body encapsulation and hemolymph clotting after injury (*Söderhäll & Cerenius, 1992*).

In this study, we aimed at testing the effects of leech infection on these (mostly immune) factors in *E. verrucosus* from Lake Baikal. We started by screening the leech biodiversity in different seasons at one chosen sampling location and checking whether those leeches could indeed feed on the amphipod hemolymph. Next, we analyzed the influence of leech infection on hemocyte concentration and phenoloxidase activity in the hemolymph of *E. verrucosus*, as well as on the amount of available glycogen recources. In search of potential highly pronounced effects of leech saliva on hemocytes, we extracted these immune cells in the primary culture from infected and uninfected individuals and compared the intensity of their aggregation around model foreign bodies. Finally, we used a bacterial strain of the genus *Pseudomonas* originally isolated from the hemolymph of *E. verrucosus* to estimate the modulating effect of leech infection on the amphipod ability to maintain hemocyte concentration in the hemolymph during the fight against bacterial infection. The choice of *Pseudomonas* was due to the high infection rate of *E. verrucosus* with the genus at this location (*Shchapova et al., 2021*) and, thus, the necessity to check for potential synergistic effects between two most frequently found parasites of the amphipod.

## MATERIALS & METHODS

### Animal sampling and handling

All experimental procedures were conducted in accordance with the EU Directive 2010/63/EU for animal experiments and the Declaration of Helsinki; the protocol of

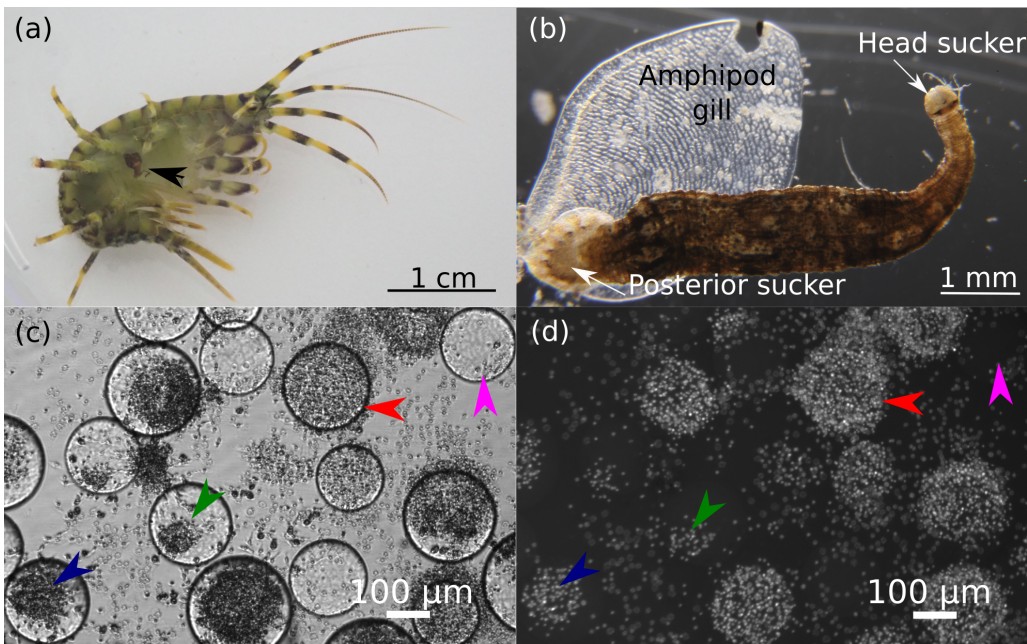

**Figure 1** **Representative photos of the research objects.** (A) Photo of amphipod *Eulimnogammarus verrucosus* with a leech attached to its gills indicated by the black arrow. (B) Microscopic photo of a leech with amphipod gill after detachment. (C, D) Hemocytes of *E. verrucosus* and stages of their encapsulation reaction to Sephadex beads after DAPI staining. c, bright field channel; d, DAPI channel of the inverted fluorescent microscope. Pink arrows, no response; green arrows, low response; dark blue arrows, intermediate response; red arrows, intense response. Figure source credit: Anna Nazarova.

the study was approved by the Animal Subjects Research Committee of the Institute of Biology at Irkutsk State University (protocol #2022/11) before the start of the experiments. Leech-free and leech-infected amphipods *Eulimnogammarus verrucosus* (Gerstfeldt, 1858) were collected by kick sampling with a hand net in Baikal littoral zone near Listvyanka village (51°52′05.5″N 104°49′47.1″E) at depths of 0–1.2 m (the animals belong to the W genetic lineage (*Drozdova et al., 2022*)). Amphipods were acclimated to the laboratory conditions in well aerated 3-L plastic aquaria at 6°C in MIR-254 incubators (Sanyo, Osaka, Japan) for at least 3 days prior to any manipulations and experiments. All found leeches were attached to the gills of amphipods (Figs. 1A, 1B).

## Identification of leech species

After samplings in October 2022, February 2023, and April 2023 and subsequent acclimation, some of the leeches were detached from amphipods and fixed in 96% ethanol for further species identification. Body width of fixed leeches was determined after photographing under a stereo microscope SPM0880 (Altami, Russia) in ImageJ software (*Rueden et al., 2017*).

Morphological analysis of fixed specimens was performed according to the standard keys (*Bauer, 1987*; *Lukin, 1976*). DNA extraction from the posterior sucker of leeches was performed using the S-sorb kit (Syntol, EX-516, Moscow, Russia). PCR amplification of

the cytochrome c oxidase subunit I (*COI*) gene fragment was performed with a $5\times$ Screen Mix (Evrogen, Moscow, Russia), the Folmer primers (LCO1490/HCO2198 (*Folmer et al., 1994*)), and the following program: 94 °C for 1 min, 30 cycles of 94 °C for 20 s, 43 °C for 2 min, and 72 °C for 1 min.

The sequencing reactions were performed in both directions using the BigDye Terminator v3.1 Cycle Sequencing Kit (Life Technologies, Waltham, MA, USA) and analyzed with a Nanophor-05 Sanger sequencer (Syntol, Moscow, Russia). Sequencing reads were basecalled and converted with the programs Mutation Surveyor v5.1 and Chromas v2.6.6. Consensus sequences were compiled with UGENE v41.0 (*Okonechnikov et al., 2012*) using the sequence from *Baicalobdella* sp. (NCBI Genbank MN854834) as the reference *COI* fragment. The obtained sequences with a length of 559 bp were deposited in the NCBI GenBank database with accession numbers OR077511–OR077525. Almost all *COI* sequences for *Baicalobdella* sp., as well as one sequence for a closely related genus *Codonobdella*, deposited in NCBI as of 5th February 2024 (*Bolbat et al., 2021*; *Utevsky & Trontelj, 2004*), were used (except for KM078844, KM078841, KM078820 and KM078810 due to their lengths shorter than 559 bp) to construct the phylogeny along with the obtained data. Several sequences of fish leeches of the genus *Piscicola* (KM095104, DQ414337, OX030972, and MH395321) were used as outgroups (*Kaygorodova et al., 2014*; *Utevsky & Trontelj, 2004*; *Cichocka et al., 2018*). Sequences were aligned with the MAFFT algorithm (*Katoh & Standley, 2013*) in the UGENE program (*Okonechnikov et al., 2012*); the alignment is available in Supplemental Information. The phylogeny was built with the IQ-Tree web server (http://www.iqtree.org/) using automatic model selection with Model Finder (*Kalyaanamoorthy et al., 2017*) and ultrafast bootstrap for assessment of the branch support values (*Hoang et al., 2018*). The resulting phylogenetic tree was visualized with iTOL (https://itol.embl.de/) (*Letunic & Bork, 2021*).

## Injection of fluorescent latex beads into amphipods and further visualization

We analyzed the ability of leeches *Baicalobdella* sp. to consume amphipod hemolymph after sampling in July 2023. For this, leeches were detached from amphipod gills with tweezers and kept in aquaria separately from hosts for ∼24 h. Next, 10 non-infected individuals of *E. verrucosus* were immobilized in an incised wet polyurethane sponge at the acclimation temperature and injected with 1 μl of saline containing about $3 \times 10^6$ latex microbeads (L3030, Sigma-Aldrich, St. Louis, MO, USA) using an IM-9B microinjector (Narishige, Tokyo, Japan). Right after the injection, the amphipods were placed in aquaria with free leeches, which attached to the new hosts within 30 min.

Four hours post-injection, we anesthetized the amphipods in clove oil suspension (50 μL of clove oil per 50 mL of Baikal water) and detached leeches and two pieces of gills from each individual for further observation under an inverted fluorescent microscope Celena S (Logos Biosystems, Republic of Korea). Prior to the visualization, the leeches were placed into sterile 1.5-mL microtubes and homogenized with 50 μL of phosphate buffered saline using a plastic pestle.

## Hemolymph extraction and characterization of hemocytes

In all experiments before the hemolymph extraction, the dorsal side of the amphipod pereon surface was always sterilized with 70% ethanol. The central hemolymph vessel was punctured with a sterile needle, and hemolymph was collected with a sterile glass capillary. The obtained hemolymph was immediately mixed 1:1 with isotonic anticoagulant solution (150 mM NaCl, 5 mM $Na_2HPO_4$, 30 mM sodium citrate, 10 mM EDTA, pH 8.0; filtered through a 0.45 µm syringe filter) on ice to avoid degranulation of granulocytes (*Shchapova et al., 2019*). Amphipod hemolymph was always extracted before the detachment of leeches.

Hemocytes were visualized using the Celena S inverted microscope (Logos Biosystems, Republic of Korea) or the Mikmed-2 upright microscope (LOMO, Russia) with an attached EOS 1200D camera (Canon, Tokyo, Japan). Total hemocyte count (THC; *i.e.*, hemocyte concentration in a certain volume) and granulocyte percentage were estimated in glass hemocytometers or disposable hemocytometers (Aptaca, Italy).

Characterization of hemocyte types was performed with a CytoFLEX flow cytometer (Beckman Coulter, Brea, CA, USA). The hemolymph of 8 non-infected amphipods *E. verrucosus* was extracted and measured for forward (allows for the discrimination of cells by size) and side scatter (gives the information about cell complexity).

## Biochemical measurements of glycogen content and phenoloxidase activity

Along with the estimation of THC, some of infected and non-infected animals collected in October 2022, February 2023 or April 2023 were used for glycogen content measurements. Glycogen along with lipids and protein content are the main resources depleting under energy demand in crustaceans (*Sánchez-Paz et al., 2006*; *Sacristán et al., 2017*). Glycogen extraction was performed as described previously (*Vereshchagina et al., 2016*) with modifications. Frozen amphipod tissues (after hemolymph extraction) were ground into a powder, mixed with the solution (0.5 mL per 100 mg of wet weight) containing 0.6 M $HClO_4$, and further homogenized in a Potter-Elvehjem tissue grinder until no visible particles remained. Next, 20 µL of the homogenate were mixed with 75 µL of 1% amyloglucosidase (10115-5G-F, Sigma-Aldrich, Germany; 5,250 U/µL) in a 0.2 M acetic acid buffer (acetic acid/sodium acetate; pH 4.8). The mix was incubated at 40 °C for two hours and then 62.5 µL of 0.6 M $HClO_4$ and 100 µL of 1 M $KHCO_3$ were added. The supernatant was centrifuged at $16000 \times g$ for 15 min. Glycogen concentration was measured with the "Glucose-Vital" kit (Vital Development, Russia): 40 µL of experimental sample was added to 190 µL of "Glucose-Vital" monoreagent and incubated at 25 ° C for 15 min. Light absorption was measured at 510 nm with a CLARIOstar Plus microplate reader (BMG Labtech, Ortenberg, Germany).

Hemolymph phenoloxidase activity was measured for amphipods sampled in May 2023. For this, hemolymph was collected as described above, mixed 1:1 with a buffer solution (150 mM NaCl, 10 mM $Na_2HPO_4$, 7 mg/mL phenylmethanesulfonyl fluoride, pH 8.0), and frozen at −80 °C. The samples were thawed at 4 °C and centrifuged for 10 min at 500 g and 4 °C to precipitate the cellular pellets. 10 µL of hemolymph extract were mixed with 40 µL of buffer solution, 280 µL of distilled water, and 40 µL of 4 mg/mL

3,4-dihydroxy-L-phenylalanine. Measurements were performed with the CLARIOstar Plus microplate reader at 490 nm (absorbance) for 40 min. The activity of phenoloxidase was calculated as the slope of the reaction curve during the linear phase and expressed in arbitrary units (*Shchapova et al., 2019*).

## Assessing the encapsulation of Sephadex beads by amphipod hemocytes in primary culture

The encapsulation reaction of hemocytes extracted from leech-infected and non-infected amphipods was quantitatively assessed using primary cell culture with added Sephadex beads as model foreign bodies (*Wu et al., 2014*). This *in vitro* approach allowed us to maintain similar concentrations of hemocytes and the beads in different replicates of the experiment, which would hardly be possible with *in vivo* experiments due to the highly variable hemocyte concentration in hemolymph.

Sephadex microbeads (G100120-50G, Sigma-Aldrich, St. Louis, MO, USA) were washed with 5 mg/mL streptomycin and 5,000 U/mL penicillin solution (1.3.18, Biolot, St. Petersburg, Russia). The bead suspension was pipetted into a sterile 96-well plate (GT204-0096DV, Minimed, Northridge, CA, USA) in a laminar flow box. Then, 100 μL of complete medium L-15 (Leibovitz medium with L-glutamine, L4386-10X1L, Sigma-Aldrich, St. Louis, MO, USA) containing 15% fetal bovine serum (FBS-HI-11A, Capricorn Scientific, Germany) was added (*Shchapova et al., 2019*) and the beads were imaged under a Mikmed-2 microscope and counted using the CountThings application (https://countthings.com). Since initially the amount of microbeads per well varied substantially, for further tests we used only the wells with approximately the same number of beads (on average $230 \pm 70$ beads per well).

Hemolymph was extracted from 10 leech-free and 12 leech-infected amphipods (collected in April 2023) as described above and pooled within each group. 10-μL aliquots were collected from the pools to estimate the hemocyte concentrations. Then, each pool of hemolymph was divided into the selected wells with microbeads with control for the equal amounts of hemocytes for the leech-free (15 wells) and leech-infected (11 wells) groups (on average, $1 \pm 0,4 \times 10^5$ cells per well).

After cell sedimentation to the well bottom, the upper layer of the suspension was collected, and 100 μL of fresh L-15 medium with L-glutamine and 15% fetal bovine serum were added; cells were kept at 6 °C (*Shchapova et al., 2019*). The hemocyte response to the Sephadex microbeads was analyzed after 24 h of incubation, and the number of microbeads with hemocyte aggregates was counted (*Mastore et al., 2015*; *Wu et al., 2014*; *Ling & Yu, 2006*) under the Celena S inverted fluorescent microscope. We categorized four stages of the encapsulation reaction: no reaction, low reaction, medium reaction, the stage showing partially encapsulated beads, and the intense reaction showing fully covered beads (Figs. 1C, 1D). The hemocyte nuclei were stained with 10 μg/mL 4′,6-diamidino-2-phenylindole (DAPI, A4099, AppliChem, Darmstadt, Germany) to visually contrast the encapsulation reaction (Fig. 1D). Cell viability was assessed by staining with 1 μg/mL propidium iodide (81845-100MG, Sigma-Adrich, St. Louis, MO, USA).

## Artificial infection of amphipods with leeches and bacteria

In order to evaluate the potential synergistic effects of infection with bacteria and leeches on amphipod immune system, we performed two 3-day-long experiments. The first (auxillary) experiment was intended to check for the possible influence of injection (sham treatment) on the studied parameters, THC and granulocyte percentage. The experiment included the initial control group and amphipods (sampling of leech-free animals in February 2024) after injection of 2.5 µL of buffered saline (150 mM NaCl, 10 mM $Na_2HPO_4$) into the central hemolymph vessel between the 5th and 6th segments with an IM-9B microinjector (Narishige, Tokyo, Japan). Animals were immobilized in an incised wet polyurethane sponge at the acclimation temperature during all injections. After 1.5 h, 1 and 3 days hemolymph was extracted from amphipods and mixed 1:1 with adjusted anticoagulant solution (150 mM NaCl, 5 mM $Na_2HPO_4$, 30 mM sodium citrate, 10 mM EDTA, 50 mM EDTA-$Na_2$, pH 8.0). This adjusted anticoagulant solution allows to fix hemocytes in the state when nuclei and granules are visible more clearly (*Skafar & Shumeiko, 2022*) and was applied to later visually distinguish granulocytes among all hemocytes. THC and granulocyte proportion were estimated under the Mikmed-2 microscope in a glass hemocytometer.

For the second (main) experiment (animal sampling in July 2023), we injected the bacterial strain *Pseudomonas* sp. H5-2 (belongs to the *P. fluorescens* species group) that was previously extracted from the hemolymph of *E. verrucosus* collected in the same location (*Shchapova et al., 2021*). For the cultivation, we used the tryptic soy broth (TSB) medium (casein peptone, dipotassium hydrogen phosphate, glucose, NaCl, and soy peptone) as suggested previously (*Robach, 2006*; (*Murali, Bhargava & Wright, 2018*). For injection into amphipods, *Pseudomonas* sp. cells were washed by centrifugation and resuspended in physiological solution in order to achieve a concentration of $10^5$ *Pseudomonas* sp. cells per 1 µL (*i.e.,* $2.5 \times 10^5$ cells per animal). After ~15–30 min, the amphipods with and without the bacterial injection were infected by leeches as described above with a 1:1 parasite-to-host ratio.

All experimental groups of the main experiment for 1.5 h and 1 day time points included 10 animals per group and showed no mortality both with and without bacterial infection (since hemolymph samplings always failed for some animals, the number of analyzed hemolymph samples had to be reduced down to seven for some groups). The first round for the 3-day time point also included 10 animals per experimental group but showed high mortality specifically for animals injected with bacteria (60% for leech-free and 50% for leech-infected), with no mortality for amphipods without injection. Since this high mortality could be an artifact of the specific injection procedure, we performed the second round of the experiment with bacterial injection into 9 animals per experimental group. Both leech-free and leech-infected animals showed no mortality during 3 days post injection, and their hemolymph was used for the tests along with the hemolymph of the animals from the first round.

## Statistical analysis

All data analyses were performed in R v.4.3.1 using built-in functions (*R Core Team, 2022*). Statistically significant differences between experimental groups were always estimated using the Mann–Whitney U test with Holm's correction for multiple comparisons. The differences were considered statistically significant at $p < 0.05$. The $p$-values to linear regression coefficients for the relation between THC and summarized leech width per host were obtained with the summary() function.

Specifically for the experiment with artificial infections with bacteria and leeches, we applied a generalized linear model (GLM) for the analysis of factor effects. The model was fitted using the glm() function with Gaussian distribution to three independent factors (time as numeric variable, absence or presence of leech, and injected bacteria) and all of their interactions. The assumptions for GLM were mostly met for the dataset: the outcome with time was acceptably linear (slightly violated specifically for THC), the residuals were always homoscedastic, and the normality assumption was slightly violated only for THC.

# RESULTS

## Infection rates and identification of leeches

We collected leech-infected amphipods *E. verrucosus* at the same site in Lake Baikal but at different times of the year. The infection rate was not estimated precisely, but it clearly varied greatly: in October 2022 94 individuals out of ∼120 examined (∼78%) were infected, in February 2023 11 out of ∼130 (∼9%) and in April 2023 12 out of ∼100 (∼12%).

We performed a morphological analysis of 35 leeches obtained from amphipods that were further used for estimation of hemocyte concentration (five leeches in October 2022, 15 leeches in February 2023 and 15 leeches in April 2023). All 35 analyzed leeches belonged to the same genus *Baicalobdella*, with most of them being representatives of the morphospecies *B. torquata*. Four leeches (sampled in February 2023) were identified as potentially belonging to the morphospecies *B. cottidarum*, but recent data indicate that *B. torquata* may have significant morphological variability (*Matveenko & Kaygorodova, 2020*; *Matveenko, 2023*), so identification of these specimens remained uncertain. In order to clarify the diversity of the leeches, we performed sequencing of the *COI* gene fragment in 15 specimens in total; all samples with ambiguous morphological identification and from October 2022 were included in the analysis, and the rest morphologically identified as *B. torquata* were chosen randomly from two samplings.

The phylogenetic tree clearly showed that all 15 leeches belonged to the same species *B. torquata* (Fig. 2); their *COI* fragments also showed low pairwise differences of no more than 1.8% (*i.e.,* 10 mutations per 559 bp). Since *E. verrucosus* was found to be infected with only one species of *Baicalobdella* locally, we had the possibility to test the physiological influence of these leeches on the amphipods.

## Leeches *B. torquata* consume amphipod hemolymph

The assumption that the leeches attached to the gills of amphipods also feed on their hemolymph is obvious, but these ectosymbionts may simply be in phoretic relationships with specifically these hosts. In order to test this assumption, we injected fluorescent

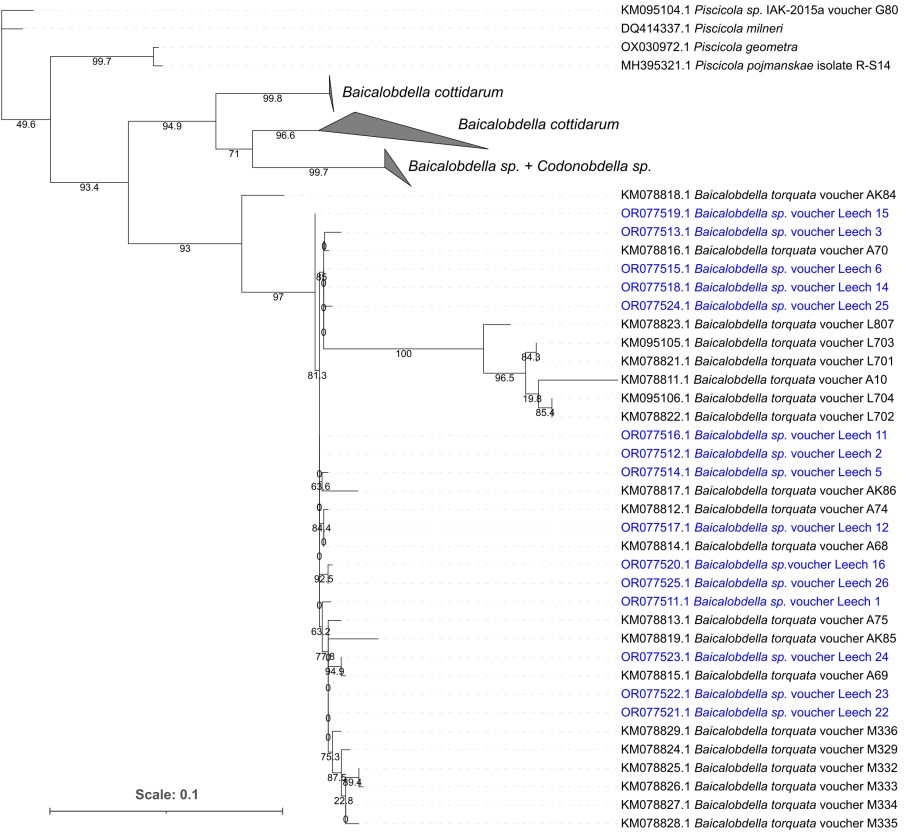

**Figure 2** **Phylogenetic tree of partial *COI* gene sequences of leech samples detached from amphipods *E. verrucosus* collected in Baikal littoral zone nearby Listvyanka village (highlighted in blue) and sequences of other closely related leeches.** The numbers next to the nodes mean percent of their ultrafast bootstrap support.

microbeads into the hemolymph of *E. verrucosus* and tracked their distribution. Five hours post-injection, the microbeads were easily observable in amphipod gills and also inside some ciliates that were found to be attached to gills (Figs. 3A, 3B). The homogenates of five out of 10 tested leech bodies also contained these fluorescent microbeads (Fig. 3C). Since the leech oral apparatus is not suitable for consumption of ciliates (*Bauer, 1987*; *Nesemann & Neubert, 1999*; *Sawyer, 1986*), our data unambiguously confirms that leeches *B. torquata* can indeed feed on the hemolymph of *E. verrucosus*.

## Characterization of amphipod hemocytes

Since hemocytes are an important component of the crustacean immune system, before further analysis we investigated their possible subdivision into populations. Flow cytometry clearly differentiated the hemocytes of *E. verrucosus* into two main groups, one with a smaller cell size and lower internal complexity and the other with a larger cell size and higher internal complexity (Fig. 4). The groups are usually called hyalinocytes and granulocytes, respectively, (*Rowley, 2016*) and can also be differentiated with conventional phase contrast microscopy. In particular, the larger size of granulocytes is evident right after

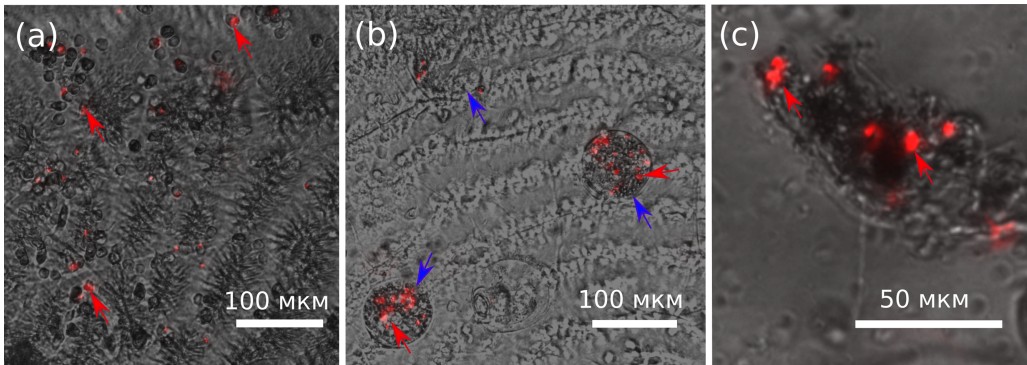

**Figure 3** **Distribution of latex microbeads 5 h after injection into the central amphipod hemolymph vessel.** (A) Amphipod gill with latex microbeads and hemocytes. (B) Ciliate cells on the surface of gills with the microbeads inside them. (C) Content of leech body with latex microbeads. The pictures are merged photos obtained in brightfield and RFP channels with the same camera settings. Red arrows, latex microbeads; blue arrows, ciliates with microbeads inside. Figure source credit: Anna Nazarova.

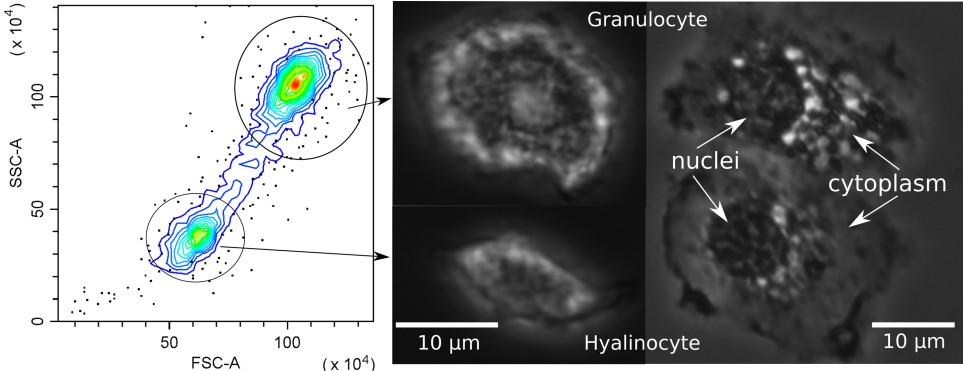

**Figure 4** **Characterization of *E. verrucosus* hemocytes using flow cytometry and microscopy.** Left panel shows the internal complexity (side scatter, SSC) against the cell size (forward scatter, FSC) of hemocyte populations (hyalinocytes and granulocytes), and other panels depict their respective phase contrast photos before (center) and after (right) attachment to a glass surface. Figure source credit: Anna Nazarova.

the sample is placed under the microscope, while the higher amount of vesicular structures in granulocytes is better visualized after attachment to the surface (Fig. 4). Additionally, we observed hemocytes with intermediate internal complexity and size between granulocytes and hyalinocytes, *i.e.,* semi-granulocytes, but their proportion was only ∼10%.

## Influence of leeches on hemocyte concentration and other parameters of amphipods in different seasons

The consumption of hemolymph by leeches may directly reduce the hemocyte concentration and phenoloxidase content in the hemolymph and indirectly reduce the available energy resources such as glycogen due to the compensation of the tissue loss. We used the amphipods collected in October 2022, February 2023 and April 2023

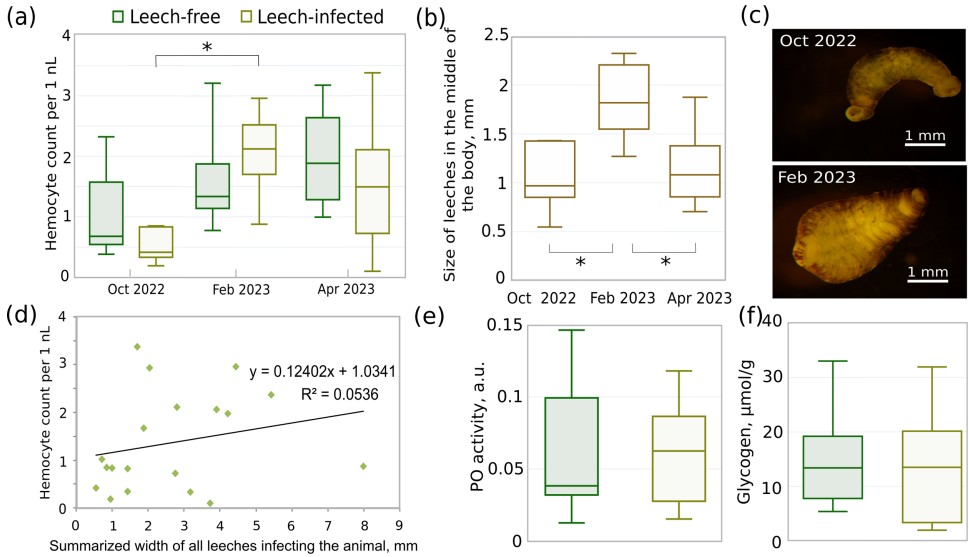

**Figure 5** Different parameters of acclimated leech-infected and non-infected (leech-free) amphipods *E. verrucosus* collected from natural environment and sizes of their leeches in different seasons. (A) Total hemocyte count (*i.e.*, hemocyte concentration) of infected and non-infected *E. verrucosus* collected in different seasons (n = 5–10). (B) Width of leeches in the middle of the body in different seasons. (C) Representative photos of leeches detached from *E. verrucosus* in different seasons. Figure source credit: Anna Nazarova. (D) Dependence of hemocyte count on summarized width of all leeches infecting the animal. The difference of the regression coefficient from zero is not statistically significant with $p = 0.34$. (E) Phenoloxidase activity in hemolymph of leech-free and leech-infected *E. verrucosus* collected in May 2023 (n = 8–10). Color legend is identical to (A). (F) Amount of glycogen in leech-free and leech-infected *E. verrucosus* collected in October 2022, February and April 2023 (n = 9–10). Color legend is identical to (A). Asterisks (*) indicate statistically significant differences with $p$-value < 0.05.

to discriminate between the effects of leech infection on two of these parameters of *E. verrucosus* in different seasons.

The median total hemocyte count (THC) of non-infected animals gradually increased from October to April by 2.8 times (Fig. 5A), but the difference between seasons was not statistically significant (all three $p$-values > 0.12). There were also no significant differences in THC between leech-infected and non-infected amphipods in these months (all three $p$-values > 0.42). However, THC for infected animals in February was significantly higher than in October ($p < 0.01$) and by median 1.5 times higher than in respective non-infected animals. This coincides with over 1.6 larger median width of leeches in February (Figs. 5B, 5C) in comparison to both October and April (both $p$-values < 0.005), while the size in the latter two months was effectively identical ($p = 0.57$). With a larger leech size the hemocyte concentration would be expected to be the lowest, but the obtained data suggested no such relation or even the opposite tendency. Since after acclimation most amphipods were infected with 2–4 leeches (11 out of 19 infected), we could not check the correlation between THC and leech size directly, but the dependence between THC and summarized leech width per host was practically absent with Spearman's correlation coefficient of 0.36 (Fig. 5D).

The analysis of glycogen content (Fig. 5F) included uniformly selected samples from October, February, and April and showed identical median values between leech-infected and non-infected amphipods ($p = 0.51$). Finally, phenoloxidase activity was measured for the separate set of *E. verrucosus* sampled in May 2023 (Fig. 5E) and indicated no statistically significant differences between infected and non-infected amphipods ($p = 0.9$).

## Cellular immune response of infected and non-infected amphipods estimated *in vitro*

Despite we did not find substantial effects of leech infection on the amounts of immune components in amphipod hemolymph, they might modulate the intensity of the host immune response through bioactive components in their saliva. For preliminary testing of this hypothesis, we chose the primary culture of amphipod hemocytes as a convenient model system and Sephadex microbeads (consisting of specifically processed dextran) as model foreign bodies. The primary hemocyte culture allows for observing the behavior of these immune cells and quantitative estimation of their reactions such as aggregation and further encapsulation of foreign bodies.

In particular, we measured the fraction of Sephadex beads encapsulated by hemocytes that were originally extracted from leech-infected and non-infected amphipods 24 h after contact with the beads. This time point was previously shown to be enough for the development of a strong immune reaction even to artificial, non-microbial foreign bodies (*Shchapova et al., 2019*). We found no difference in the intensity of the immune reaction between the experimental groups since the proportions of fully encapsulated (~6%) and partially encapsulated microbeads (~93%) were equal (both *p*-values > 0.07) for hemocytes from infected and non-infected amphipods (Fig. 6A). Some of the beads were not encapsulated at all, and there was a high mortality of hemocytes around Sephadex microbeads in contrast to free hemocytes, as indicated by propidium iodide staining (Fig. 6B).

## Changes in hemocyte concentration and composition after injection of bacteria and artificial leech infection

Finally, in order to evaluate the potential synergistic effects of leeches and other immunity-related factors we experimentally tested the influence of leeches on the ability of amphipods to deal with bacterial infection. First, we examined the potential effects of the injection procedure on the chosen parameters. Both THC and the fraction of granulocytes among all hemocytes (Figs. 7A, 7B) demonstrated no statistically significant changes during the 3 days after injection of physiological solution in comparison to amphipods without any injections (all six *p*-values > 0.32 in comparisons to the respective control groups).

Next, we performed the experiment with (i) an artificial infection of leech-free amphipods with the *Pseudomonas* sp. strain originally extracted from hemolymph of the same species and (ii) an artificial infection with leeches. The order of the procedures was motivated by the high infection rate of *E. verrucosus* with *Pseudomonas* (*Shchapova et al., 2021*) and the variability in infection rate with leeches (see above). The number of injected bacterial cells was comparable to the number of circulating hemocytes in the

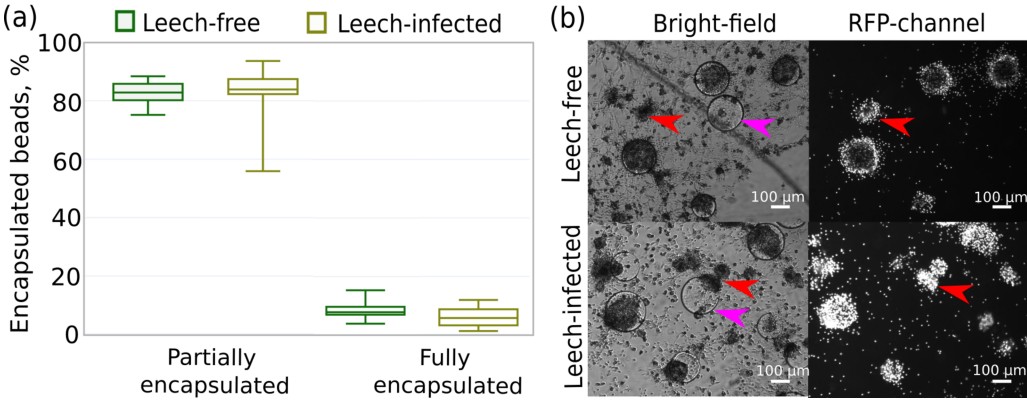

**Figure 6 Intensity of the cellular immune response of hemocytes extracted in primary culture from leech-infected and non-infected (leech-free) amphipods *E. verrucosus*.** (A) Fractions of Sephadex microbeads partially and fully encapsulated by hemocytes after 24 h of contact. (B) Example photos of microbeads' encapsulation in hemocyte primary culture, propidium iodide staining in RFP channel. Pink arrows—Sephadex microbeads, red arrows—aggregates of hemocytes. Photos in RFP channel were obtained at the same camera settings in different groups, but time of staining could be different.

animal hemolymph to model a significant microbial infection. Some of the amphipods were then infected with one leech per animal (Fig. 7; Table 1).

The mortality during the three-day experiment was mostly low, and it was never higher for leech-infected animals than for leech-free ones. The generalized linear model indicated that infection with leeches itself and time after the infections had no statistically significant effects on both the concentration of hemocytes in amphipod hemolymph and the fraction of granulocytes among them, while the injection of bacteria clearly leads to a statistically significant decrease in hemocyte concentration by ~2,800 cells per μl on average and an increase in granulocyte proportion by ~16% (Table 1). Interestingly, the interaction of bacterial injection and leech infection, in contrast, resulted in a statistically significant decrease in the proportion of granulocytes by 12% but caused no statistically significant changes in hemocyte concentration (Table 1). Other interactions between factors, even being statistically significant in the case of granulocyte percentage, did not exceed 0.5% in absolute value in the estimated effect. However, pairwise comparisons between leech-free and leech-infected animals showed no statistically significant differences between any experimental groups not only in hemocyte concentration but also in granulocyte fraction during the whole experiment (all twelve *p*-values > 0.09; Fig. 7).

## DISCUSSION

Our research group focuses on the environmental physiology of the amphipods endemic to Lake Baikal, and almost exclusively the previously published experiments were made with amphipods without visible leech infection (*Drozdova et al., 2019*; *Jakob et al., 2016*; *Axenov-Gribanov et al., 2016*; *Bedulina et al., 2013*) since the infected individuals were considered as potentially weakened. Here we questioned this assumption.

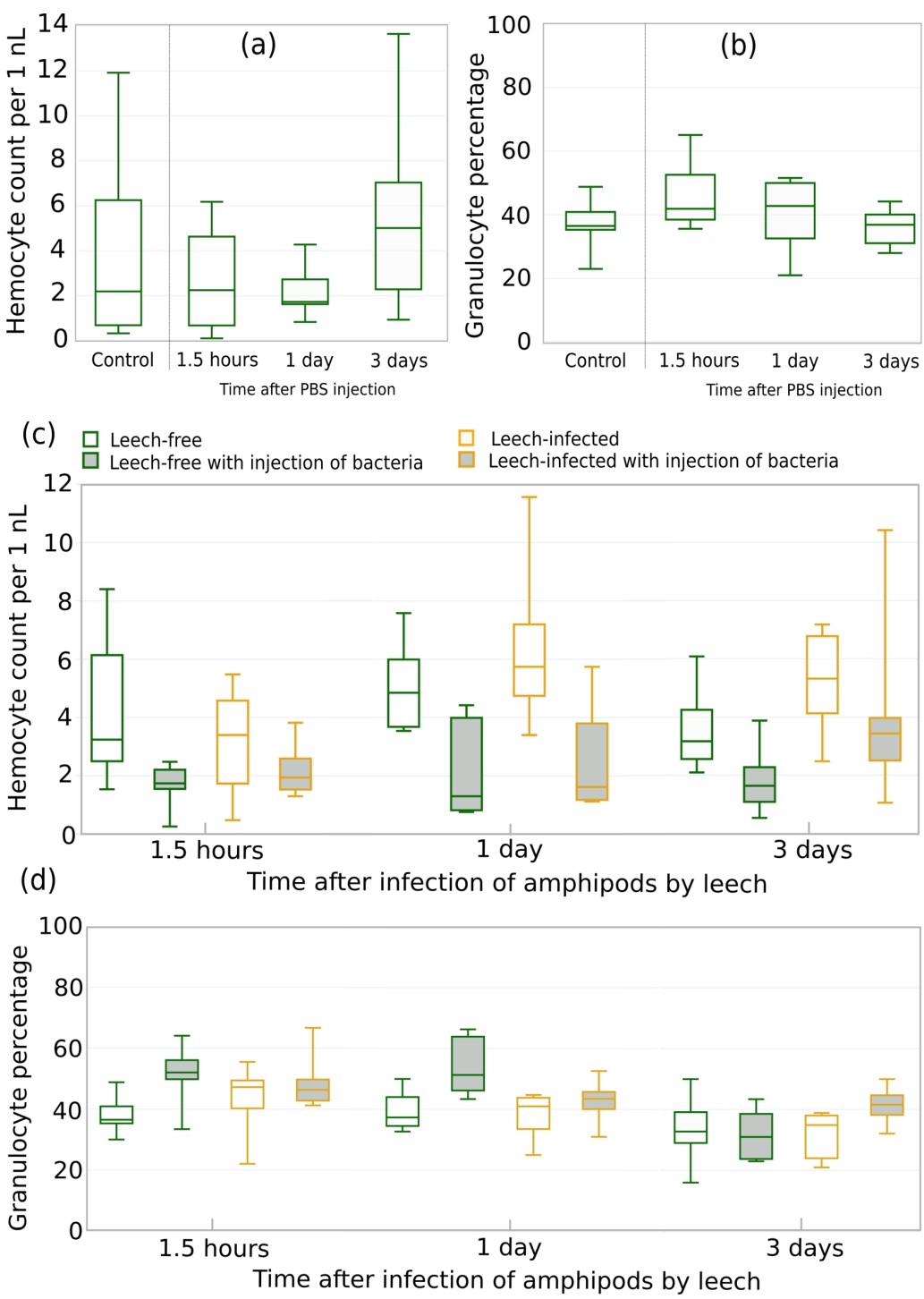

**Figure 7** **Reaction of amphipod immune cells to bacterial injection and artificial leech infection.** (A) Total hemocyte count (*i.e.,* hemocyte concentration; n = 7–8) and (B) granulocyte fraction among all hemocytes of amphipods injected with saline buffer (n = 5–8). (C) Total hemocyte count and (D) granulocyte fraction among all hemocytes (n = 7–13) after bacterial injection and artificial leech infection. The legend is identical for (C) and (D). Injection of bacteria to amphipod central hemolymph vessel was performed about 15 min before leech infection.

**Table 1 Output of generalized linear model with the Gaussian distribution fitted to total hemocyte count and granulocyte percentage in leech-free and artificially leech-infected amphipods with and without bacterial injection (see Figs. 7C, 7D).** All interactions between factors were allowed, but the results only for three independent factors and their statistically significant interactions with substantial effect estimates are depicted here. *, $p < 0.05$; ***, $p < 0.001$.

| Variable | Total hemocyte count | | Granulocyte percentage | |
|---|---|---|---|---|
| | Estimate, cells/μl | P-value | Estimate, % | P-value |
| Time | −15.1 | 0.248 | −0.061 | 0.284 |
| Leech | −514.1 | 0.537 | 4.149 | 0.256 |
| Bacteria | −2,846.1 | <0.001*** | 16.457 | <0.001*** |
| Leech:Bacteria | 603.3 | 0.601 | −12.223 | 0.019* |

Sequencing of leeches from *E. verrucosus* collected in three independent sampling campaigns clearly demonstrated that the parasites in the used sampling location belong to the same species *B. torquata* (Fig. 2), and thus their influence on these amphipods can be studied without preliminary species identification. Since the phylogeny and diversity of leeches in Lake Baikal are still being revised (*Kaygorodova & Sorokovikova, 2014*; *Bolbat, Bukin & Kaygorodova, 2022*; *Kaygorodova, 2015*; *Kaygorodova & Matveenko, 2023*) and specifically *B. torquata* is a complex of cryptic species (*Matveenko & Kaygorodova, 2020*), the genetic lineage analyzed in our study (we observed no cryptic diversity in the chosen location) can later be assigned a different species name. Our tests also showed that the leeches can consume amphipod hemolymph (Fig. 3), and thus the effects of the infection on amphipod physiology are worth studying. However, within the 4-hour experiment, only a half of the artificially attached leeches consumed hemolymph, and during our samplings in nature leeches were always attached to gills with their posterior sucker, which indicates that these parasites do not consume hemolymph constantly.

The studies investigating the host-symbiont relationships of amphipods besides classical life-history traits commonly use such techniques as histological analysis, spectrophotometry, metagenomics, PCR, and microscopy, while such an important component of the immune system as hemocytes is rarely mentioned (*Bojko & Ovcharenko, 2019*). *Rigaud & Moret (2003)* studied phenoloxidase activity of *Gammarus pulex* (Linnaeus, 1758) and *G. roeselii* Gervais, 1835 and found a correlation between infection by acanthocephalans and a decrease in the enzyme activity. Another freshwater amphipod *G. fossarum* (Koch, 1835) was used as a test organism to investigate potential pollutant-parasite interactions for infection with larvae of the acanthocephalan species *Polymorphus minutus* (Zeder, 1800). Phenoloxidase activity, glycogen, and lipid concentrations significantly increased in infected *G. fossarum* individuals (*Rothe et al., 2022*). In the other case, acanthocephalan infection was associated with a reduction of phenoloxidase activity and hemocyte concentration (*Cornet et al., 2009*).

Here we were looking for any substantial effects of leech infection on hemocytes and other related parameters of amphipods. Most studies on crustacean hemocytes have been performed for decapods and revealed three main types of these immune cells with different morphology and functions: hyalinocytes (hyaline cell), semi-granulocytes (semi-granular

cells), and granulocytes (granular cells) (*Rowley, 2016*). Such information for amphipods is less abundant. Using light and electron microscopy, the following hemocyte types were found in the body of the amphipod *G. setosus* (Dementieva, 1931): granulocytes, adipohemocytes, plasmatocytes, and rare prohemocytes (*Steele & MacPherson, 1981*). In the hemolymph of the amphipod *G. pulex*, four types of circulating cells were identified with microscopy and histochemical staining: hyalocytes I (cells with a transparent cytoplasm), hyalocytes II (cells with a slightly basophilic cytoplasm), granulocytes, and adipohemocytes (cells with a large nucleus surrounded by granules) (*Schroder, Doicescu & Arcuș, 2017*). In the case of *Parhyale hawaiensis* (Dana, 1853), it was shown that hemolymph contained three typical types of hemocytes: granulocytes, semi-granulocytes and hyalinocytes with semi-granulocytes being rare (*dos Santos et al., 2023*). Our research on *E. verrucosus* seems to be the first or among the first studies checking amphipod hemocyte diversity with flow cytometry, which demonstrated the prevalence of two types of hemocytes, granulocytes, and hyalinocytes, while intermediate semi-granulocytes were found to be relatively rare (Fig. 4).

We used leech-free and leech-infected amphipods from the same samplings and of similar size in order to compare the concentration of hemocytes in hemolymph, phenoloxidase activity and glycogen content and found no influence of leeches on these parameters (Fig. 5). Thus, our data suggest that hemolymph consumption by leech is either negligible or the loss of hemocytes and phenoloxidase is compensated by the host. In the last case, there should be an energetic burden of the infection, but the observed similar glycogen contents in infected and non-infected animals do not support this hypothesis. However, we cannot fully exclude the deleterious effects of *B. torquata* on *E. verrucosus* since other energy resources such as lipids and proteins could be consumed (*Sánchez-Paz et al., 2006*; *Sacristán et al., 2017*) in this amphipod species and those were not determined. Interestingly, the median hemocyte concentration of non-infected animals varied greatly (yet without statistically significant differences) with sampling campaigns and could be partially influenced by the reproduction season, which starts in autumn for *E. verrucosus*. The infection rates were also very different in different months (dropped from ∼80% to ∼10% from October to April), which indirectly indicates that the same individual of this species can be infected with different leeches multiple times during their lifespan of about five years. We also observed no leeches on several initially infected individuals after acclimation (exact numbers were not recorded), which partially supports this conclusion. A previous transcriptomic study indicated that even *E. verrucosus* without visible leech infection can bear the parasites, so the mentioned values might be an underestimate (*Drozdova et al., 2019*).

Next, we checked for the potential influence of leech saliva on the intensity of reaction to artificial foreign bodies in the primary hemocyte culture. This *in vitro* approach allowed us to maintain the concentrations of hemocytes and the model foreign bodies for more stable quantitative analysis but as a drawback all humoral components of hemolymph were diluted for ∼12 times during extraction into the primary culture from leech-free and leech-infected animals. Such a design could reveal only strong or long-term effects of saliva components on amphipod hemocytes (such as changes in protein expression), and we observed no difference between the groups (Fig. 6). Thus, here we could not fully

exclude the possible minor effects of *B. torquata* saliva on the intensity of cellular immune response in amphipods of Lake Baikal. Additionally, since these leeches do not consume hemolymph constantly, the effects of their saliva could already be alleviated with time.

Finally, we checked for potential synergistic interaction of leeches with an artificial bacterial infection and found no or even a slight antagonistic interaction, as indicated by the estimates of granulocyte fraction among all hemocytes (Figs. 7C, 7D; Table 1). Although the main experiment did not include sham treatment as a separate group, the supporting experiment (Figs. 7A, 7B) indicated no effects of saline injection on the studied parameters. In the main experiment artificial infection with leech itself did not influence hemocyte concentration or granulocyte percentage in the amphipod hemolymph at all, while injection of bacteria in saline clearly decreased the first and increased the second parameter (Figs. 7C, 7D; Table 1). The decrease in THC was expected from a number of studies (*Sung, Hwang & Tasi, 2000*; *Sarathi et al., 2007*; *Ji, Yao & Wang, 2011*; *Gao et al., 2023*). The increase in the fraction of granulocytes among all hemocytes probably reflects the high mortality of hyalinocytes during the immune response to bacteria, but a possible discharge of granulocytes from some tissues also cannot be excluded. The antagonistic interaction of leech infection with bacterial injection specifically in the case of the granulocyte fraction among all hemocytes might be speculatively explained by a potential decrease in the concentration of bacteria due to hemolymph consumption by the leech, but this effect clearly demands further exploration.

An unexpected finding of our research was the discovery of numerous parasitic ciliates on the gills of *E. verrucosus* that clearly consumed amphipod hemolymph (Fig. 3B). It is known that ciliates of the family *Lagenophryidae* can attach to the gills of *E. verrucosus* (*Mayén-Estrada & Clamp, 2016*). However, their potential influence on the amphipods is a subject for separate research.

Overall, our study revealed no substantial influence of leeches *B. torquata* on the amphipods *E. verrucosus* from Lake Baikal. However, some influence cannot be fully excluded since after sampling from nature we did not check all important energy resources, while the laboratory experiments were only mid-term and included just one parasite per individual. Therefore, the amphipods infected with *B. torquata* should still be treated carefully but can be included in at least some types of ecophysiological experiments. In certain seasons high infection rates can significantly complicate collecting strictly non-infected amphipods, while permanent checking for the infection is a laborious and time-consuming process. Thus, using leech-infected *E. verrucosus* in the experiments intended for glycogen measurements or tests with primary hemocyte cultures can speed up those studies.

## ACKNOWLEDGEMENTS

The flow cytometry measurements were obtained with the equipment kindly provided by the Helicon Company (Moscow, Russia).

### Funding

This research was supported by the Russian Science Foundation (project #23-14-00165). The funder had no role in study design, data collection and analysis, decision to publish, or preparation of the manuscript.

### Grant Disclosures

The following grant information was disclosed by the authors:
The Russian Science Foundation: #23-14-00165.

### Competing Interests

The authors declare there are no competing interests.

### Author Contributions

- Anna Nazarova conceived and designed the experiments, performed the experiments, analyzed the data, prepared figures and/or tables, authored or reviewed drafts of the article, and approved the final draft.
- Andrei Mutin performed the experiments, prepared figures and/or tables, and approved the final draft.
- Denis Skafar conceived and designed the experiments, performed the experiments, prepared figures and/or tables, and approved the final draft.
- Nadezhda Bolbat performed the experiments, analyzed the data, authored or reviewed drafts of the article, and approved the final draft.
- Sofya Sedova performed the experiments, prepared figures and/or tables, and approved the final draft.
- Polina Chupalova performed the experiments, prepared figures and/or tables, and approved the final draft.
- Vasiliy Pomazkin performed the experiments, authored or reviewed drafts of the article, and approved the final draft.
- Polina Drozdova analyzed the data, authored or reviewed drafts of the article, and approved the final draft.
- Anton Gurkov conceived and designed the experiments, analyzed the data, authored or reviewed drafts of the article, and approved the final draft.
- Maxim Timofeyev analyzed the data, authored or reviewed drafts of the article, and approved the final draft.

### Ethics

The following information was supplied relating to ethical approvals (i.e., approving body and any reference numbers):

Animal Subjects Research Committee of the Institute of Biology at Irkutsk State University.

## DNA Deposition

The following information was supplied regarding the deposition of DNA sequences:

The COI sequences of leech samples are available at the NCBI GenBank database: OR077511–OR077525.

https://www.ncbi.nlm.nih.gov/nuccore/OR077525.1/.

## Data Availability

The raw data for flow cytometry of hemocytes on Fig. 3 and boxplots on Figs. 5, 6 and 7, as well as the sequence alignment used for building tree on Fig. 2 are available in the Supplementary Files.

They are also available at Zenodo: Nazarova, A., Mutin, A., Denis, S., Bolbat, N., Sofya, S., Chupalova, P., Pomazkin, V., Drozdova, P., Anton, G., & Timofeyev, M. (2024). Leeches Baicalobdella torquata feed on hemolymph but have a low effect on the cellular immune response of amphipod Eulimnogammarus verrucosus from Lake Baikal [Data set]. Zenodo. https://doi.org/10.5281/zenodo.10777326.

## Supplemental Information

Supplemental information for this article can be found online at http://dx.doi.org/10.7717/peerj.17348#supplemental-information.

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
