# Peer review of "Leeches Baicalobdella torquata feed on hemolymph but have a low effect on the cellular immune response of amphipod Eulimnogammarus verrucosus from Lake Baikal"

_PeerJ, doi:10.7717/peerj.17348_

## Round 0.1 · original submission · Major Revisions

Three recognized experts have assessed your manuscript and identified a number of issues that make the manuscript unacceptable in its present form. The most important aspects are:

(1) deficits in the structure of the manuscript, in particular the confusion of statements assigned to the introduction, material and methods, results and discussion chapters. In the three reviews you will find very prominent examples that require restructuring your manuscript. The aims of the study should be provided at the end of the introduction, a justification of the methods used already in the methods chapter and not only in the results chapter as currently done.

(2) experimental design: the lack of a sham-infection has to be discussed (see. reviewer 1)

(3) a more cautious discussion: The reviewers do not agree with your conclusion that the parasite has neglectable effects on the host, given the limited number of effects you have analysed in your study. They recommend greater caution in interpreting the results and I fully agree. Also, you should focus more closely on your results in the discussion and avoid speculations lacking any empirical basis.

(4) statistical analyses: Reviewer 3 re-assessed your raw data and came to different results compared to your analyses. This issue has to be solved. Please re-visit your statistical analyses and comment on any differences you may find compared to the calculations provided by reviewer 3

On the other hand, the reviewers underlined the unique position and great importance of your study, so I hope that their criticisms will allow you to carry out a substantial revision of the manuscript, which is a precondition for its acceptance.

Reviewer 1 ·

Basic reporting

The text is well written and english language is clear and understandable. Rare exceptions are listed below.
The introduction gives a good overview on the topic and clarifies the research question and novelty including relevant references.
Structure is according to peerJ guidelines. There are some issues concerning the mixture of Results/introduction/discussion. See specific points under “general comments”. Raw data is made available. Figures are of good quality and all figures are relevant for the paper.

Experimental design

The study fits to the scope of peerJ. The content is novel, interesting and relevant as there is only scarce knowledge on leech parasitism on amphipods. Further, physiological/immunological effects of these parasites on their host are provide which generally adds to our knowledge on the effects of parasites on their hosts.
A number of tests were conducted to clarify the parasite feeding rate and the effect on the immune status and immune response which provides a good basis for the further conclusions. For the effect on energy reserves only glycogen was assessed. Here it should be made clear that other parameters like lipids or protein levels might be affected by the leech feeding activity.
The bacteria-infection experiment is problematic as no sham-infection (injection with medium without bacteria) was performed, especially as high mortality was observed in the injected group after 3d. Nevertheless, conclusions can be drawn from this experiment but the methodological problem has to be discussed.

Validity of the findings

The obtained results are valid and novel showing an unexpectedly low impact of parasitism by leeches on several parameters of the host. Nevertheless, I believe the authors should be careful to conclude that there is no effect of the parasite at all, as other parameters (e.g. lipids, protein) or long term effects (e.g. reduced fecundity) might be impaired (particularly considering the large size of the parasite compared to the host).

Additional comments

In the introduction, you might consider citing this paper in addition/alternatively to Sures 2006:
Grabner, D., Rothe, L. E., & Sures, B. (2023). Parasites and Pollutants: Effects of Multiple Stressors on Aquatic Organisms. Environ Toxicol Chem, 42(9), 1946-1959. https://doi.org/10.1002/etc.5689

L32-34. “Artificial infection…” This sentence is complicated and hard to understand. Please rephrase.
L167: “thawed” instead of “melted”
L186: Is there a reference for the use of Sephadex beads to assess hemocyte encapsulation?
L264: delete “us”; “belonged”
L265: Do you mean “dominance” instead of “prevalence”?
L274: “belonged”
L275-279: Should be moved to discussion.
L282-283: This should be made clear in the introduction as research question. Here you should report only the results.
L290: Is it possible to observe a biting wound of the leech at the gills after removing it? That would further support this finding.
L316-317/L323-324: This would be discussion.
L329-330: This again is a discussion-statement. Also, I do not fully agree. Other parameters (lipids, protein) could be impaired and where not measures. The conclusion that the leeches do not have deleterious effects cannot be drawn based on your findings. The leeches are quite large compared to their host so it is hard to believe that they don’t have any effect. This should be discussed. (please also correct the final statement in the abstract accordingly).
L337-338: You talk about “primary hemocyte culture” but the cultivation was not described in the methods.
L363: better “infection” instead of “outburst”; see also Abstract
L388: As mentioned above, you cannot fully exclude an effect of the parasite according to your results. Also, I would still recommend to use uninfected individuals (or consider infection as a factor) for any kind of experiments, particularly for such an easily recognizable parasite.

Caption Fig. 2: Please explain the meaning of the red colouration of Baicalobdella sp.
Caption Fig. 5 “leech-infected and non-infected”

·

Basic reporting

Nazarova and colleagues evaluated whether the leeches observed on the Baikal amphipod Eulimnogammarus verrucosus feed on the hemolymph of the host. After establishing that leeches indeed feed on host hemolymph, the authors conducted a series of experiments to assess whether the cellular immune response of leech infected amphipods differ from non-infected amphipods. The authors report no significant differences. These findings are important and represent the first report of this kind of amphipod-leech relationship, uncovering yet another facet of the peculiarity of Baikal’s biodiversity.

I enjoyed reading this interesting manuscript. The presented evidence is convincing, and the methodology is described in detail and seems appropriate (I am not a specialist in physiology). The text is well written and easy to understand. I made a few small remarks on the attached pdf file.

My only main remark is that I got lost while reading the Methods section as there was rarely any justification as to why this or that particular method was employed. It was not until I got to the Results that I understood why these various methods were employed. The explanations in the Results were good and justified well the usage of the methods. However, I think these explanations could be added already in the Methods to make reading smother and to better follow the thread of the manuscript. The overall description of the Methods seems very detailed and sufficient so that the study can be reproduced.

Experimental design

no comment

Validity of the findings

no comment

Reviewer 3 ·

Basic reporting

The manuscript should be re-structured, mainly to present the aims of the different experiments at the beginning of the article (see comments)

Experimental design

Scientific questions should be more defined and justified. Experiment-by-experiment methods are fine (but not always justified).

Validity of the findings

This is a pioneer study describing the interaction between leeches and amphipods. In such it is interesting. However, to me, basic descriptions of this interaction are lacking (prevalence, average number of parasite/host, phenotypic effects on hosts, etc.) I found that the discussion should be focussed on the results, and not on speculative opinions.

Additional comments

This work is describing the effects of Baicalobdella sp. leeches found on the gills of a Baikal amphipod. Broadly, authors suggest that leeches indeed feed on the hemolymph of their host. However, authors suggest that these leeches have a weak detrimental effect on their host, because: the infection did not affect immune parameters (hemocyte concentration and phenoloxidase activity), nor glycogen content. In the same way, they claim that artificial infection with leeches did not influence parameters of bacterial infections.
I have many comments on this manuscript that would considerably improve it.
1- First, as a general comment, the reader does not understand by reading the introduction (especially the end) what are the detailed aim of the study. We are discovering these aims scattered in the result sections, generally coming from nowhere. I therefore suggest that these aims should be transferred either at the end of the introduction or in the material and methods. For example, not only the immune parameters were studied, but also (1) identification of leeches (2) if they really ingest hemolymph (are they parasites or "just" phoretic animals) (3) if energy (glycogen) contents is impacted... Therefore, L. 282-285 , L. 309-311, L. 333-339, L. 357-359 should be moved around here.

2- A second general comment is that, for such a pioneer study on leech infection in amphipods, I would have begun by obtaining basic life-history data on infected versus uninfected hosts such as: sex, size, number of females carrying eggs, egg counting, etc., before thinking about immunity.

Now going gradually in the manuscript:
3- L. 131: The aim of this experiment is not provided here. It is in the result section, which is a bit late for an easy understanding. So please move L. 282-285 in this Mat&Met section.
4- L. 161: why Glycogen? (why not lipids for example?)
5- L. 209: about Fig. 1 C, D. On this picture, we can see that hemocyte aggregates without beads. So how can you ensure that the hemocytes on beads are not just aggregates superposing with the beads, but not attached to the beads? You did not rinsed the bead, don't you?
6- L. 214: at this stage, it has not been explained why you are infecting amphipods with bacteria. The link with the leeches is not clear. Again, this should be justified before.
7- L. 217: Do you know if the pseudomonas are pathogenic to amphipods? Can you provide predictions in cases of surinfections? Above all, I do not see the interest of doing bacterial infections before the leech infection. The aim of the paper is to see the effect of leeches, right? So you should have infect with leech, and then infect with bacteria, to see if leech is favouring the infection by other parasite or pathogen. Please justify the order of this experimental infection.
8- L. 224-225: did you controlled that all leeches attached with their mouth part and not their posterior part? Else making blood sample make no sense...
9- L. 260-262: Apparently, the prevalence was not measured properly. Therefore, a percentage of infected animals cannot be provided !!!! Please give prevalence (re-measure it) or remove this sentence. The same is true L. 262-263 : either you provide data or you say nothing. But you cannot write “There were from one to nine leeches per animal with 2-4 parasites found on most individuals”… How many amphipods were investigated? How many were uninfected? How many infected? Etc…
10- L. 265. I do not understand the word “prevalence” in this sentence.
11- L. 278. In the tree, the name Codonobdella is given. In the text, the name is Baicalobdella. This is very confusing, please be consistent.
12- L. 288-289. How can you be sure thata leeches are not predatory in your species? It is a new species, so can you provide some insurance that the oral apparatus is similar to other known leeches?
13- L. 315: Are size statistically different?
14- L. 317. why should larger leeches switch the host? If you are describing a new infection, how can you provide this life-cycle precision? In addition this is discussion, not result.
15- L. 322. You cannot say that: the relationship is not significant. By the way, The R2 on Figure 5D is 0.05, not 0.5.
16- L. 323-324. Please move to discussion.
17- L. 325-328. Please provide statistical test supporting these sentences. There is apparently a strong effect of season on THC (see figure. You did not analyzed it this way, but variation between season is significant!) So PO or glycogen can also vary between season, and the effect of leeches on these parameters can vary... I would therefore suggest prudent interpretation of these results.
18- L. 346-348. See comment #5. Fig 6 is showing aggregates not located on beads... so we can imagine that the cell culture is leading to hemocyte aggregation without beads. I would suggest injecting beads within the heamocel and then re-sample to see the degree of bead encapsulation.
19- L. 349-353. This should be moved to discussion, and I find this explanation very speculative. About the last sentence : due to the small sample size for all these experiments, for sure you cannot exclude an effect !
20- L. 367-371. I found strange from the figures that there was no effect of leech in interaction with bacteria or time. I therefore re-run the stats from the data provided, and I found quite different results than the authors. See statistical models below. For hemocyte count I used a Poisson distribution (it is a count), corrected for overdispersion (“quasi-Poisson”). The third model is the one minimizing the AICc (the “optimal” model). For the % granulocytes, I used GLM with normal distributions. I ran model with time as either a continuous variable or categorical variable (both are equally justified, I guess, because there are only time-points).

GLM with Poison distribution, with time as a category, corrected for overdispersion
Response : Hem count
N = 95
Whole model
Model -Log-likelihood Khi square likelihood ratio Df P
Difference 29.7689218 59.5378 11 <.0001*
Complete 40.8612337
Reduced 70.6301556

Quality of adjustment Khi square Df P Overdispersion
Pearson 77157.09 83 <.0001* 929.6034
Déviance 75038.90 83 <.0001*

AICc
112.2163

Effects
Source Df Khi square likelihood ratio P
time 2 1.1426784 0.5648
leech 1 10.361068 0.0013
time*leech 2 2.7945709 0.2473
bacteria 1 25.276017 <.0001
time*bacteria 2 4.0462434 0.1322
leech*bacteria 1 4.775211 0.0289
time*leech*bacteria 2 5.9777719 0.0503


GLM with Poison distribution, with time as a continuous factor, corrected for overdispersion
Response : Hem count
N = 95
Whole model
Model -Log-likelihood Khi square likelihood ratio Df P
Difference 20.8497733 41.6995 7 <.0001*
Complete 42.2540265
Reduced 63.1037998

Quality of adjustment Khi square Df P Overdispersion
Pearson 90521.48 87 <.0001* 1040.477
Déviance 86998.08 87 <.0001*

AICc
104.6257

Effects
Source Df Khi square likelihood ratio P
time 1 0.0012219 0.9721
leech 1 12.226116 0.0005
bacteria 1 24.491463 <.0001
time*leech 1 1.2470987 0.2641
time*bacteria 1 0.0190474 0.8902
Leech*bacteria 1 4.3706901 0.0366
time*leech*bacteria 1 0.7595653 0.3835

GLM with Poison distribution, minimizing the AICc, corrected for overdispersion
Response : Hem count
N = 95
Whole model
Model -Log-likelihood Khi square likelihood ratio Df P
Difference 19.9830328 39.9661 3 <.0001*
Complete 44.4887587
Reduced 64.4717915

Quality of adjustment Khi square Df P Overdispersion
Pearson 92674.35 91 <.0001* 1018.399
Déviance 89684.07 91 <.0001*

AICc
99.6517

Effects
Source Df Khi square likelihood ratio P
leech 1 12.342827 0.0004
bacteria 1 26.121086 <.0001
leech*bacteria 1 4.7902147 0.0286


GLM with normal distribution. Time as category
Response : proportion Granu
N = 92
Whole model
Model -Log-likelihood Khi square likelihood ratio Df P
Difference 28.4517692 56.9035 11 <.0001*
Complete -107.75192
Reduced -79.300154

AICc
-184.8372*

Effects
Source Df Khi square likelihood ratio P
time 2 30.101321 <.0001
leech 1 0.0470034 0.8284
time*leech 2 6.8166884 0.0331
bacteria 1 21.804316 <.0001
time*bacteria 2 2.9723312 0.2262
leech*bacteria 1 0.6674786 0.4139
time*leech*bacteria 2 9.7021923 0.0078


GLM with normal distribution. Time as continuous factor
Response : proportion Granu
N = 92
Whole model
Model -Log-likelihood Khi square likelihood ratio Df P
Différence 24.90112 49.8022 7 <.0001*
Complete -104.20127
reduced -79.300154

AICc
-188.2074*

Effects
Source Df Khi square likelihood ratio P
time 1 27.399696 <.0001
leech 1 0.0010064 0.9747
bacteria 1 18.945883 <.0001
time*leech 1 1.215207 0.2703
time*bacteria 1 2.4715262 0.1159
leech*bacteria 1 0.2406614 0.6237
time*leech*bacteria 1 8.1406284 0.0043

21- L. 379-382. This statement is too strong from the results !. There is apparently at least an interaction. However, again, what is the question? Here the answer seems to be "after bacterial infection, the hemocytes count decrease; but after a leech surinfection seems the hemocyte count is reversed”. I would have been much more interested by answering the question "Is a leech infection favours bacterial infection (or is it preventing bacterial infection?), and what is the consequence on immunity?" here we have the answer to the question "in case of bacterial infection, what is the effect of leech infection on immunity?" Why is it in this order? Do authors have indications that most gammarids are infected by bacteria, and more rarely by leeches? (in other words, this experiment has been made as if bacteria infection was very common, and leech are always secondary infections. This could be OK, but this should be said!). Many authors have shown that the order of infection is important in the outcome of multiple infections, so that, a minima, the experiment should be done the other way round. (BTW Why should there be synergy?)
22- L. 397-399. Can you please quantify this? This is important.
23- L. 401. Also studies on life-history traits are commonly made, which is the basis, but which was not done here... it is lacking.
24- L. 432-435. No data are provided about this in this manuscript. Please either provide quantitative data or do not discuss that point. Please discuss on your results, not on "possible" observations.
25- L. 440-441. what do you mean by "more natural" ? What is the link between absence of immune response and the fact that it is "natural"? Infection of human being by lyme disease, for instance, is "natural", but induce strong immune response !!!
26- L. 443-446. Here should be discussed the fact that leeches are not always sucking blood (at least we have no data indicating that each leech was attached by its mouth in all experiments).
27- L. 448-449. From my own calculations the effect is stronger than reported here.

---

## Round 0.2 · Minor Revisions

Thank you for the thorough revision of the manuscripts which helped to solve most of the problerms identified by the reviewers in the first revision round. As identified by reviewer 1, there are still some pending issues with the language.
I look forward to your revised manuscript.

Reviewer 1 ·

Basic reporting

Minor comments on language:
L29-31: I suggest rephrasing e.g.: „…no effect on immune parameters such as haemocyte concentration, or phenoloxidase activity and also did not affect glycogen content.”
L75: „detectable“
L95: “does not mention”
L104: delete “roughly”
L314: Suggest rephrasing, e.g.: “We collected leech-infected E. verrucosus amphipods at the same site in Lake Baikal, but at different times of the year.”
L318-322: I suggest condensing this a bit, e.g.: “We performed a morphological analysis for 35 leeches that were further used for estimation of hemocyte concentration (5 leeches in October 2022, 15 leeches in February and 15 leeches in April 2023). All 35 analyzed leeches belonged to the same genus Baicalobdella, with most of them being representatives of the morphospecies B. torquata.”
L345: delete “amphipods”
L357: “hyalinocytes”
L427-429: maybe rephrase: “Interestingly, the interaction between bacterial injection and leech infection, in contrast, led to a statistically significant decrease in the proportion of granulocytes...”
L430-431: suggest rephrasing: “Interestingly, the interaction of bacterial injection and leech infection resulted in a statistically…”
L517: “Although the main experiment…”

Those are just some examples. I suggest that the authors do a final round of checking the text for clarity.

Experimental design

All fine in the revised version.

Validity of the findings

L437-441: Maybe clarify the problem. E.g. that it is time consuming to check for leeches before the experiment, or that not enough uninfected individuals can be found. Otherwise it is not clear why it is not better to just continue using uninfected amphipods to be on the safe side for physiology experiments.

Additional comments

The autors did a great job revising the manuscript and addressing the points raised. Besides a few comments on language I can recommend the manuscript for publication in PeerJ!

·

Basic reporting

The authors have adequately addressed my initial concerns. The manuscript is now clearer and better organized.

Experimental design

Adding additional control groups has improved the experimental design. The experiments are generally better justified.

Validity of the findings

The findings have improved since the authors managed to precisely identify the leach species.

---

## Round 0.3 · accepted · Accept

Thank you for the revision of the manuscript. I hereby certify that you have adequately taken into account the comments and improved the manuscript accordingly. Based on my assessment as an Academic Editor, your manuscript is now ready for publication. I am very happy that you have selected PeerJ for the publication of your study.

---

## Author Rebuttal · Round 0.3

**Editor comments:**

Thank you for the thorough revision of the manuscripts which helped to solve most of the problerms identified by the reviewers in the first revision round. As identified by reviewer 1, there are still some pending issues with the language.
I look forward to your revised manuscript.

Authors: Dear Prof. Oehlmann, thank you very much for consideration of our study and the provided suggestions. We now incorporated the comments of Reviewer 1 and re-checked the whole text for grammar mistakes and clarity as suggested.

Thank you very much again for consideration of our manuscript.

**Reviewer 1**

**Basic reporting**

Minor comments on language:

L29-31: I suggest rephrasing e.g.: „…no effect on immune parameters such as haemocyte concentration, or phenoloxidase activity and also did not affect glycogen content."

L75: „detectable"

L95: "does not mention"

L104: delete "roughly"

L314: Suggest rephrasing, e.g.: "We collected leech-infected E. verrucosus amphipods at the same site in Lake Baikal, but at different times of the year."

Authors: Dear Reviewer 1, thank you very much for critically reading our manuscript and the warm comments. Those pieces of text were corrected as suggested.

L318-322: I suggest condensing this a bit, e.g.: "We performed a morphological analysis for 35 leeches that were further used for estimation of hemocyte concentration (5 leeches in October 2022, 15 leeches in February and 15 leeches in April 2023). All 35 analyzed leeches belonged to the same genus Baicalobdella, with most of them being representatives of the morphospecies B. torquata."

Authors: Corrected in the following way: «We performed a morphological analysis for 35 leeches obtained from amphipods that were further used for estimation of hemocyte concentration (5 leeches in October 2022, 15 leeches in February and 15 leeches in April 2023). All 35 analyzed leeches belonged to the same genus *Baicalobdella*, with most of them being representatives of the morphospecies *B. torquata*».

L345: delete "amphipods"

L357: "hyalinocytes"

L427-429: maybe rephrase: "Interestingly, the interaction between bacterial injection and leech infection, in contrast, led to a statistically significant decrease in the proportion of granulocytes..."

L430-431: suggest rephrasing: "Interestingly, the interaction of bacterial injection and leech infection resulted in a statistically…"

L517: "Although the main experiment…"

Authors: Corrected as suggested.

Those are just some examples. I suggest that the authors do a final round of checking the text for clarity.

Authors: We re-checked the whole manuscript for typos, grammar mistakes and clarity.

**Experimental design**

All fine in the revised version.

**Validity of the findings**

L437-441: Maybe clarify the problem. E.g. that it is time consuming to check for leeches before the experiment, or that not enough uninfected individuals can be found. Otherwise it is not clear why it is not better to just continue using uninfected amphipods to be on the safe side for physiology experiments.

Authors: This is certainly a relevant criticism. We added this motivation at the very end of Discussion (in order to keep the logical line at the beginning as it was): «Therefore, the amphipods infected with *B. torquata* should still be treated carefully but can be included into at least some types of ecophysiological experiments. In certain seasons high infection rates can significantly complicate collecting strictly non-infected amphipods, while permanent checking for the infection is a laborious and time-consuming process. Thus, using leech-infected *E. verrucosus* in the experiments intended for glycogen measurements or tests with primary hemocyte cultures can speed up those studies».

**Additional comments**

The autors did a great job revising the manuscript and addressing the points raised. Besides a few comments on language I can recommend the manuscript for publication in PeerJ!

Authors: Thank you very much again for your work and the provided suggestions!